# Preparation of MoFs-Derived Cobalt Oxide/Carbon Nanotubes Composites for High-Performance Asymmetric Supercapacitor

**DOI:** 10.3390/molecules28073177

**Published:** 2023-04-03

**Authors:** Caiqin Yang, Weiwei Li, Xiaowei Liu, Xiumei Song, Hongpeng Li, Lichao Tan

**Affiliations:** 1Institute of Carbon Neutrality, Zhejiang Wanli University, Ningbo 315100, China; 2School of Materials Science and Chemical Engineering, Harbin University of Science and Technology, Harbin 150040, China; 3Chilwee Power Co., Ltd., No. 18 Chengnan Road, Huaxi Industrial Zone, Changxing 313100, China; 4College of Mechanical Engineering, Yangzhou University, Yangzhou 225127, China; 5State Key Laboratory of Clean Energy Utilization, School of Materials Science and Engineering, Zhejiang University, Hangzhou 310027, China

**Keywords:** asymmetric supercapacitor, electrode materials, MOFs derivatives, cobalt compounds, carbon nanotubes

## Abstract

Metal–organic frameworks (MOFs)-derived metallic oxide compounds exhibit a tunable structure and intriguing activity and have received intensive investigation in recent years. Herein, this work reports metal–organic frameworks (MOFs)-derived cobalt oxide/carbon nanotubes (MWCNTx@Co_3_O_4_) composites by calcining the MWCNTx@ZIF-67 precursor in one step. The morphology and structure of the composite were investigated by scanning electron microscope (SEM) and transmission electron microscope (TEM) characterization. The compositions and valence states of the compounds were characterized by X-ray diffraction (XRD) and X-ray photoelectron spectroscopy (XPS). Benefiting from the structurally stable MOFs-derived porous cobalt oxide frameworks and the homogeneous conductive carbon nanotubes, the synthesized MWCNTx@Co_3_O_4_ composites display a maximum specific capacitance of 206.89 F·g^−1^ at 1.0 A·g^−1^. In addition, the specific capacitance of the MWCNT_3_@Co_3_O_4_//activated carbon (AC) asymmetric capacitor reaches 50 F·g^−1^, and has an excellent electrochemical performance. These results suggest that the MWCNTx@Co_3_O_4_ composites can be a potential candidate for electrochemical energy storage devices.

## 1. Introduction

The desire to reduce society’s dependence on fossil fuels has made the exploration of new energy and high energy utilization efficiency one of the most important issues faced by national governments in the 21st century. Energy storage, which is an intermediate step toward the efficient utilization of energy, has attracted large-scale concern and increasing research interest [1]. Among various emerging energy storage technologies, supercapacitors (SCs), also known as electrochemical capacitors (ECs), are a new concept of energy storage devices that was developed at the forefront of the available environmentally friendly electrochemical energy storage systems. According to the energy storage mechanisms, there are two types of supercapacitors: electrical double-layer capacitors (EDLCs) and pseudocapacitors. The core of supercapacitors is the electrode material, which directly dominates the performance of energy storage [2,3,4,5,6,7,8,9,10,11,12,13,14,15,16,17,18].

In recent years, pseudocapacitors based on transition metal oxides/hydroxides with variable valence exhibit a higher specific capacitance. Moreover, transition metal oxides and hydroxides have attracted extensive interest due to their high theoretical capacity, low cost, environmental friendliness and great flexibility in morphology and structures, such as Co_3_O_4_, NiO, Fe_3_O_4_ and MnO_2_ for pseudocapacitor electrodes, which achieve ultrahigh values of electrode capacitance [19,20,21,22,23,24,25,26]. Among the transition metal oxides, due to its wide potential range, low cost and environmental friendliness, Co_3_O_4_ has attracted much attention for its good catalysis, adsorptive behavior and electrochromic applications [27,28,29]. However, Co_3_O_4_ systems are limited by several adverse factors, such as a poor electrical conductivity and volume changes during the cycle process, a poor reversibility and a degradation in capacity at higher current densities [30]. Therefore, the rational design of sophisticated electrode structures that accelerate the charge transfer and enable the sufficient exposure of the surface active site is highly essential for Co_3_O_4_-based materials to achieve high-performance supercapacitors.

In addition, the main electrode materials, carbon materials, have been deeply studied with the widening of the application areas of supercapacitors because of their high surface area, chemical stability, electrical conductivity and low cost [31,32,33,34,35,36]. Carbon nanotubes, graphene, template carbon and carbide-derived carbon have been widely used in EDLCs.

In our study, we demonstrate a simple and economic strategy for the preparation of an MWCNTx@Co_3_O_4_ composite. This composite has a large surface area, which enables the effective exposure of the active site for the Faradaic redox reaction and obviously promotes the charge transfer process, and the diamond dodecahedron structure of the MWCNTx@Co_3_O_4_ composite can promote the deep ion diffusion among electrode materials. As a result, the MWCNTx@Co_3_O_4_ composite exhibits a high specific capacitance of 206.89 F·g^−1^ at a current density of 1.0 A·g^−1^. Moreover, the specific capacitance of the MWCNT_3_@Co_3_O_4_//AC asymmetric capacitor reaches 50 F·g^−1^, and has an excellent electrochemical performance. Therefore, the MWCNTx@Co_3_O_4_ composite can be considered as promising materials for high-performance electrochemical capacitors.

## 2. Results and Discussion

The SEM of the MWCNTx@Co_3_O_4_ sample is shown in Figure 1. As can be seen from Figure 1a–d, Co_3_O_4_ in MWCNTx@Co_3_O_4_ samples all show a rhomboidal dodecahedron morphology, indicating that Co_3_O_4_ retains the morphology of ZIF-67 and is not damaged by high temperature. The dispersion of MWCNT in MWCNT_1_@Co_3_O_4_ and MWCNT_2_@Co_3_O_4_ samples was also observed because the content of carbon nanotubes was less than that of Co_3_O_4_. On the other hand, it is possible that the surface oxidation of MWCNT has oxygen-containing functional groups that repel each other under electrostatic action so that the agglomeration of MWCNT is greatly reduced [37]. As can be seen from Figure 1d, because the MWCNT content is the highest, the whole MWCNT_4_@Co_3_O_4_ sample is wrapped by MWCNT, resulting in the Co_3_O_4_ polyhedron structure not being obvious. It even hindered the growth of the polyhedral structure, and the morphology distribution was uneven, which may be determined by the growth mechanism of ZIF-67 on the MWCNT surface. However, the distribution of MWCNTS in MWCNT_3_@Co_3_O_4_ is more reasonable, as shown in Figure 1c. The length and diameter of MWCNTs is 500 nm and 20 nm, respectively. The Co_3_O_4_ rhomboidal dodecahedron is clear and uniform in size, and the MWCNT is evenly dispersed. This phenomenon may be caused by the growth mechanism of MWCNTx@ZIF-67: under the action of a PVP dispersant, MWCNT is uniformly dispersed and can only adsorb a certain amount of Co^2+^ on the surface, while the content of ZIF-67 crystal is fixed and, after calcination, uniformly dispersed MWCNT_3_@Co_3_O_4_ [38] is obtained. According to the above characterization analysis, the morphology of the MWCNT_3_@Co_3_O_4_ composite is the most uniform.

The microstructure and composite structure of MWCNT_3_@Co_3_O_4_ were further studied by TEM characterization, as shown in Figure 2. Figure 2a,b show low and high-magnification TEM images of MWCNT_3_@Co_3_O_4_. It can be seen that the Co_3_O_4_ surface is connected to MWCNT. This composite structure can improve the conductivity of Co_3_O_4_, thereby enhancing its electrochemical performance. The size of cobalt oxide in MWCNT_3_@Co_3_O_4_ is between 120–350 nm, consistent with the size captured in the SEM image. Figure 2c is the high-resolution transmission electron microscope image of the MWCNT_3_@Co_3_O_4_ sample. Lattice fringes of different widths were obtained by analyzing the lattice spacing, and were 0.239 nm (the (311) crystal plane of Co_3_O_4_ nanoparticles) and 0.29 nm (the (220) crystal plane of Co_3_O_4_ nanoparticles), respectively. The polycrystalline properties of MWCNT_3_@Co_3_O_4_ composites are described. Figure 2d is the selected electron diffraction diagram of the MWCNT_3_@Co_3_O_4_ composite material, which indicates four bright concentric diffraction rings. The radius of the diffraction ring corresponds to the crystal surfaces of the cubic Co_3_O_4_ nanoparticles (511), (400), (200) and (311), which are consistent with the XRD results, and also indicate the polycrystalline property of the material. In summary, the successful preparation of MWCNT_3_@Co_3_O_4_ composite materials is indicated by the above experimental results.

The EDS analysis method was used to obtain the element composition and distribution map of the MWCNT_3_@Co_3_O_4_ composite. It can be observed from Figure 3 that the content of the C element is the highest, which is mainly attributed to the conversion of the organic skeleton into a C source after calcination in ZIF and the addition of an appropriate amount of MWCNT. It can also be observed that Co and O elements are evenly distributed on the surface of the polyhedron structure, which confirms the existence of Co_3_O_4_, which is the basic condition for obtaining a good pseudocapacitance performance.

Table 1 shows the composite ratio of MWCNT@Co_3_O_4_. As can be seen from the table, with an increase in the amount of MWCNT added, the content of the C element in the composites gradually increased from 21.23% to 28.50%, whereas the content of the Co element in the composites gradually decreased from 50.65% to 44.32%. This confirms the addition of MWCNT. Through the above analysis, the successful synthesis of the MWCNT_x_@Co_3_O_4_ composite materials is confirmed.

Figure 4a,b are XRD patterns of the preparation precursor MWCNTx@ZIF-67 and the MWCNTx@Co_3_O_4_ sample after calcination in air. As can be seen from Figure 4a, MWCNTx@ZIF-67 has obvious 7.2° (011), 10.4° (002), 12.7° (112), 14.7° (022), 16.4° (013), 18.1° (222), 22.1° (114), 24.5° (233), 26.5° (134), 29.6° (044), 31.3° (244), and characteristic peaks of 32.5° (235) and 43.1° (100) that are consistent with those reported in the literature [39], which proves the successful synthesis of the MWCNTx@ZIF-67 precursor. The XRD results of MWCNTx@Co_3_O_4_ are shown in Figure 4b. As can be seen from the picture, MWCNT_1_@Co_3_O_4_, MWCNT_2_@Co_3_O_4_, MWCNT_3_@Co_3_O_4_ and MWCNT_4_@Co_3_O_4_ have (511), (400), (200), (111), (311) and (422) crystal plane diffraction peaks consistent with Co_3_O_4_ (JCPDS card number 74-2120). It is proved that the MWCNTx@Co_3_O_4_ sample was successfully obtained after calcination. In addition, it can be seen from the XRD pattern of the MWCNTx@Co_3_O_4_ composite that the carbon-derived peak (002) is not obvious, which may be caused by the relatively large the diffraction peak intensity of Co_3_O_4_ on the one hand and the low crystallinity of carbon after calcining MWCNTx@ZIF-67 on the other hand.

The chemical composition and corresponding valence states of the MWCNT_3_@Co_3_O_4_ composite were studied by means of XPS characterization. All XPS maps were standardized with reference to the C 1s peak (284.6 eV). Figure 5a shows the full XPS spectrum of the MWCNT_3_@Co_3_O_4_ composite, which mainly contains C, O and Co elements. As can be seen from the Co 2p map in Figure 5b, 779.9 eV and 796.6 eV correspond to Co 2p3/2 and Co 2p1/2, respectively, which are two peaks of Co^2+^ and Co^3+^, and the experimental results are consistent with the literature [40]. As can be seen from the C 1s map in Figure 5c, there are three obvious characteristic peaks at 284.3, 285.8 and 286.4 eV. The peaks at 284.3 eV and 285.8 eV belong to C-C and C=C, respectively, whereas the peaks at 284.6 eV may belong to Co-O-C [41]. Figure 5d shows the O1s map, which mainly contains three peaks of 533.5 eV, 531.5 eV and 530.0 eV, among which the peaks of 530.0 eV and 531.5 eV belong to lattice oxygen species and surface-adsorbed oxygen [42]. These results further confirm the successful synthesis of composite materials, and are further consistent with XRD and EDS results.

Figure 6 shows the pore structure and specific surface area of the MWCNT_x_@Co_3_O_4_ composites. As can be seen from Figure 6a, the adsorption/desorption isotherm of MWCNT_x_@Co_3_O_4_ slowly rises at first, and, when it reaches a certain pressure, it rises sharply, resulting in a vertical tail ring and a hysteresis ring. This feature indicates that MWCNT_x_@Co_3_O_4_ composite materials have microporous, mesoporous and macroporous structures. The specific surface areas of MWCNT_1_@Co_3_O_4_, MWCNT_2_@Co_3_O_4_, MWCNT_3_@Co_3_O_4_ and MWCNT_4_@Co_3_O_4_ are 104.20, 66.44, 99.56 and 136.97 m^2^ g^−1^, respectively. In addition, the pore diameter distribution curve of MWCNT_x_@Co_3_O_4_ is shown in Figure 6b. The pore volumes of MWCNT_1_@Co_3_O_4_, MWCNT_2_@Co_3_O_4_, MWCNT_3_@Co_3_O_4_ and MWCNT_4_@Co_3_O_4_ were 3.39, 3.79, 3.06 and 3.30 nm, respectively. It can be well seen in the figure that the pore size distribution of MWCNT_3_@Co_3_O_4_ is mainly mesoporous, mainly due to the insertion of MWCNTs into Co_3_O_4_, resulting in a larger pore size of the overall material. Furthermore, the mesoporous structure is conducive to ion diffusion, thereby improving the electrochemical performance of the material. Therefore, the obtained electrode material should have a good electrochemical performance.

Figure 7a shows the cyclic voltammetry curves of composites with different carbon nanotube contents at 20 mV·s^−1^. As can be seen, when the scanning speed is 20 mV·s^−1^, the CV curve area of the MWCNT_3_@Co_3_O_4_ composite material is the largest, indicating that the addition of appropriate carbon nanotubes can improve the specific capacitance of the electrode. This result can be attributed to the synergistic effect of cobalt tetroxide and carbon nanotubes. MWCNT_1_@Co_3_O_4_, MWCNT_2_@Co_3_O_4_ and MWCNT_4_@Co_3_O_4_ decreased, which may be caused by a too high or too low MWCNT content and the overall electrical conductivity of the material. In addition, obvious redox peaks were observed, which were mainly due to different cobalt redox states, such as Co^4+^/Co^3+^ and Co^3+^/Co^2+^. The main reaction mechanism is shown in (1) and (2) [43]:Co_3_O_4_ + OH^−^ + H_2_O ↔ 3CoOOH+e^−^
(1)
CoOOH + OH^−^ ↔ CoO_2_+H_2_O+e^−^
(2)

Figure 7b shows the GCD curve of the MWCNTx@Co_3_O_4_ composite material, which was tested at a current density of 1 A·g^−1^. It is observed that the curve is almost triangular in symmetry, and there is no particularly rapid drop in voltage, indicating that the synthesized MWCNT_1_@Co_3_O_4_, MWCNT_2_@Co_3_O_4_, MWCNT_3_@Co_3_O_4_ and MWCNT_4_@Co_3_O_4_ materials have particularly good electrochemical reversibility. While the GCD curves of the MWCNT_4_@Co_3_O_4_ composite is different from others, its charge time is longer from its discharge time. This is mainly due to the higher resistance, which is consistent with the result of Figure 7f. When applying the same voltage in the MWCNT_4_@Co_3_O_4_ composite, the current is unable to charge quickly, which leads to the GCD curve of the sample MWCNT_4_@Co_3_O_4_ composite being asymmetrical. The discharge time of MWCNT_3_@Co_3_O_4_ is longer than that of MWCNT_1_@Co_3_O_4_, MWCNT_2_@Co_3_O_4_ and MWCNT_4_@Co_3_O_4_. Therefore, among the four materials prepared, the MWCNT_3_@Co_3_O_4_ composite has the largest specific capacitance, whereas the MWCNT_1_@Co_3_O_4_ composite material has the smallest specific capacitance. It is also comparable to other Co_3_O_4_-based materials as listed in Table 2.

Figure 7c shows the cyclic voltammetry curves of MWCNT_3_@Co_3_O_4_ at sweep speeds of 5, 10, 20, 50 and 100 mV·s^−1^. It is observed that the CV curve of the electrode material is similar with an increase in the sweep speed, indicating that the MWCNT_3_@Co_3_O_4_ electrode material has good performance. Figure 7d shows the GCD curves of MWCNTx@Co_3_O_4_ composites at 1, 2, 3, 4 and 5 A·g^−1^ current densities. When the applied current density is 1 A·g^−1^, the MWCNT_3_@Co_3_O_4_ composite has the longest discharge time and its specific capacitance is 206.89 F·g^−1^, which is higher than MWCNT_1_@Co_3_O_4_, MWCNT_2_@Co_3_O_4_ and MWCNT_4_@Co_3_O_4_ (132.73 F·g^−1^, 194.64 F·g^−1^, 89.36 F·g^−1^, respectively). It is shown that the composite has an excellent electrochemical performance. Figure 7e shows the specific capacitance as a function of current density. As the specific current density increases, the specific capacitance decreases due to the limitation of diffusion on the electrode surface [50]. The MWCNT_3_@Co_3_O_4_ electrode shows a specific capacitance of 191.67 F·g^−1^ at 5 A·g^−1^, and the specific capacity of MWCNT_3_@Co_3_O_4_ is 3.2 times higher than that of MWCNT_4_@Co_3_O_4_ and 2 times higher than that of MWCNT_1_@Co_3_O_4_ at the same current density. By adding an optimal proportion of MWCNT, the capacity of MWCNT@Co_3_O_4_ is increased, thus promoting the charge-transfer ions on the surface.

Figure 7f shows the electrochemical impedance spectroscopy (EIS) of the MWCNTx@Co_3_O_4_ composite. It can be observed that the linear slope of the composite is the highest in the low-frequency region MWCNT_3_@Co_3_O_4_, which indicates that the ion diffusion resistance of MWCNT_3_@Co_3_O_4_ is smaller than that of MWCNT_1_@Co_3_O_4_, MWCNT_2_@Co_3_O_4_ and MWCNT_4_@Co_3_O_4_ electrodes. In the high-frequency region, the MWCNT_3_@Co_3_O_4_ electrode has the smallest semicircle diameter, indicating that its charge transfer resistance is smaller than that of MWCNT_1_@Co_3_O_4_, MWCNT_2_@Co_3_O_4_ and MWCNT_4_@Co_3_O_4_, indicating that the charge transfer rate of MWCNT_3_@Co_3_O_4_ is faster. It is also further explained that the MWCNT_3_@Co_3_O_4_ electrode material has better electrical conductivity.

Figure 8 shows the cycle performance experiment conducted on MWCNT_3_@Co_3_O_4_ with a current density of 1 A·g^−1^. As can be seen, the maintenance rate of the specific capacitance of MWCNT_3_@Co_3_O_4_ is 78.9% after 1000 cycles of charge and discharge, which indicates that the MWCNT_3_@Co_3_O_4_ composite electrode has a good cycle life.

To better explore the practical application of the electrode material, the MWCNT_3_@Co_3_O_4_//AC asymmetric supercapacitor was assembled for the electrochemical test. Figure 9a shows the cyclic voltammetry curves of the MWCNT_3_@Co_3_O_4_//AC capacitor at 5, 10, 20, 50 and 100 mV·s^−1^. It can be seen from the figure that the CV curve presents the characteristics of double electric layers and pseudocapacitors, which are the conditions for the capacitor to have a good electrochemical performance. Figure 9b shows the GCD curve under different current densities. The mass specific capacitance of the capacitor under 1, 2, 3, 4 and 5 A·g^−1^ current densities is calculated as 50.0, 46.25, 34.31, 30.0 and 25.0 F·g^−1^, respectively.

Figure 9c shows the relationship between specific current and specific capacitance. It can be seen that the capacitor still has a 57.1% specific capacitance maintenance rate at a current density of 5 A·g^−1^, indicating that the capacitor has an excellent multiplier performance. Figure 9d shows the performance of the device after 1000 cycles. At a 1 A·g^−1^ current density, the initial capacitance remains at 87.2% after 1000 times of charging and discharging, indicating that the capacitor has a good cycle life. This is mainly attributed to the polyhedral structure and high porosity of MWCNT_3_@Co_3_O_4_ composites, which reduces the damage of electrode materials during charging and discharging.

Figure 10 shows the relationship between energy density and power density. Under current densities of 1 A·g^−1^, 2 A·g^−1^, 3 A·g^−1^, 4 A·g^−1^ and 5 A·g^−1^, the power densities of the device are 800, 1600, 2400, 3200 and 4000 W·kg^−1^, respectively. The corresponding energy densities are 17.78, 16.44, 12.20, 10.67 and 8.89 Wh·kg^−1^, respectively.

## 3. Experimental Section

This section may be divided by subheadings. It should provide a concise and precise description of the experimental results and their interpretation, as well as the experimental conclusions that can be drawn.

### 3.1. Materials

All of the chemical reagents in this experiment were of analytical purity and directly used without any further purification.

### 3.2. Materials Characterization

XRD analysis was performed by a powder X-ray diffraction system (XRD-6100, Rigaku, Tokyo, Japan) equipped with Cu Kα radiation (λ = 0.15406 nm) to determine crystalline structures of the obtained samples. The XPS measurements were performed by a Thermo ESCALAB 250Xi spectrometer (USA) with monochromated Al Kα radiation (hγ = 1486.6 eV). All XPS spectra were calibrated with respect to the C 1s peak at 284.6 eV. The morphology and microstructure of the samples were characterized by field emission scanning electron microscopy (FE-SEM) (JSM-6480A, JEOL, Tokyo, Japan) and a TEM (JEM-2000FX, Electronics Corporation, Tokyo, Japan). The BET (Brunauer-Emmett-Teller) and pore size distrubtion of the samples were characterized by N_2_ adsorption Nitrogen and desorption test (TriStar II3flex, Mike, Scottsdale, AZ, USA).

### 3.3. Preparation of Materials

Preparation of MWCNTx@ZIF-67 precursor: first, different masses of MWCNT (99.9%, Xianfeng Nanotechnology Co., Ltd., Nanjing, China) were poured into polyethylene pyrrolidone (analytically pure, Shanghai McLean Biochemical Technology Co., Ltd., China) and methanol solution (99.7%, Shanghai Aladdin Co., Ltd., Shanghai, China), ultrasonic was carried out for 30 min, a certain amount of cobalt nitrate (analytically pure, Shanghai Aladdin Biochemical Technology Co., Ltd., Shanghai, China) was added and then magnetic stirring was performed for 10 min. In addition, 2-methylimidazole (≥98.0%, Shanghai McLean Biochemical Technology Co., Ltd., Shanghai, China) was added to the above solution, stirring for 1 h, and then the prepared solution was left to stand at room temperature for 24 h. After centrifugation and washing, the black precipitate was dried at 60 °C for 12 h and the precursor MWCNTx@ZIF-67 was obtained. The weights of MWCNT were 0.0203 g, 0.0415 g, 0.0608 g and 0.1510 g, respectively. Therefore, the resultant MWCNT@ZIF-67 numbers are MWCNT_1_@ZIF-67, MWCNT_2_@ZIF-67, MWCNT_3_@ZIF-67 and MWCNT_4_@ZIF-67, respectively.

Preparation of MWCNTx@Co_3_O_4_ composite: the synthesized precursors (MWCNT_1_@ZIF-67, MWCNT_2_@ZIF-67 and MWCNT_3_@ZIF-67, MWCNT_4_@ZIF-67) were annealed in air at 350 °C for 2 h at a heating rate of 4 °C·min^−1^. The obtained samples were labeled MWCNT_1_@Co_3_O_4_, MWCNT_2_@Co_3_O_4_, MWCNT_3_@Co_3_O_4_ and MWCNT_4_@Co_3_O_4_.

### 3.4. Fabrication of Supercapacitor Electrode

(1)Preparation of Single Electrode

All single electrodes in this paper were prepared in the following way: firstly, certain amounts of active substance and acetylene black (Superconducting K90, Ron reagent, Harbin, China) were weighed, dissolved in 5 mL of absolute ethanol and ultrasonicated for 30 min. Then, a certain amount of PTFE (60.0 wt%, Shanghai Aladdin Biochemical Technology Co., Ltd., Shanghai, China) lotion (the mass ratio of active substance, acetylene black and PTFE lotion (5wt%) was 8:1:1) was dropped in proportion, ultrasonicated for 10 min and then dried in a constant temperature blast-drying oven at 60 °C for 12 h. Finally, the obtained black sample was scraped onto one side of foam nickel of 1 × 1 cm, pressed with a tablet press and compacted evenly. By weighing the total mass of foam nickel before and after scraping, the total mass of active substance can be calculated.

(2)Assembly of Asymmetric Supercapacitors

In order to evaluate the practical application of the prepared electrode material, the electrode material and AC prepared in this article were used to prepare a single electrode in the same manner as in (1). Subsequently, when assembling the supercapacitor, they were assembled on the positive electrode and the negative electrode of the supercapacitor, respectively. At the same time, the positive and negative electrodes were separated by a membrane, and an asymmetric supercapacitor was assembled. The electrolyte was prepared as a 3.0 M KOH (≥85.0%, Shanghai Aladdin Biochemical Technology Co., Ltd., Shanghai, China) solution.

### 3.5. Electrochemical Characterization

Electrochemical workstation (CHI 760E) was used to observe the electrochemical performance of the MWCNTx@Co_3_O_4_ composite electrode in a three-electrode installation. In this test, platinum was used as a counter electrode, Ag/AgCl as a reference electrode, MWCNTx@Co_3_O_4_ composites as a working electrode and a solution of 3 M KOH as electrolyte. Cyclic voltammetry (CV) was performed between 0 V and 0.6 V at various scan rates (from 5 to 100 mV·s^−1^). Galvanostatic discharge texts were measured in a voltage window from 0 V to 0.55 V at various current densities. EIS measurements were carried out with a 5 mV sinusoidal voltage in a frequency from 100 kHz to 0.01 Hz.

## 4. Conclusions

In summary, a diamond-dodecahedron-structured MWCNT_x_@Co_3_O_4_ composite was successfully fabricated using a smart approach. With an increase in the MWCNT content, the electrochemical properties of the MWCNT_X_@Co_3_O_4_ composite first increased and then decreased. Benefiting from the unique structure, high specific surface area and reasonable pore size distribution, the as-obtained MWCNT_3_@Co_3_O_4_ composite exhibits satisfactory capacitive behavior: 206.89 F·g^−1^ at a current density of 1 A·g^−1^; an excellent cycling stability of 87.2% capacitance retention over 1000 continuous cycles. An asymmetric supercapacitor cell was fabricated through MWCNTx@Co_3_O_4_ and AC as a positive and negative electrode, respectively. The cell can deliver a high energy density of 17.78 Wh·kg^−1^ at a power density of 800 W·kg^−1^. Our results suggest that the MWCNTx@Co_3_O_4_ composite can be used in actual high-power devices. The preparation strategy offers a facile and variable route to rationally design and prepare cobalt oxide electrode materials for a variety of applications in energy storage and conversion, catalysis and environmental treatment.

## Figures and Tables

**Figure 1 molecules-28-03177-f001:**
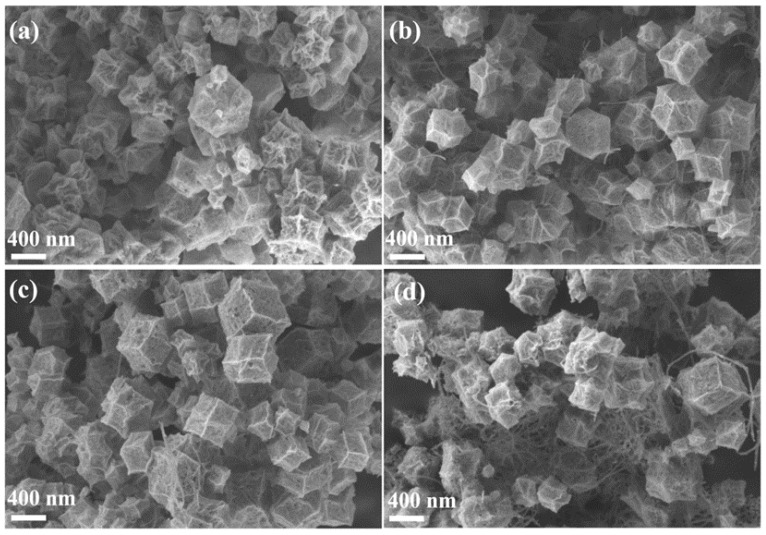
Different MWCNT additions, MWCNTx@Co_3_O_4_. SEM diagram of nanocomposites: (**a**) MWCNT_1_@Co_3_O_4_; (**b**) MWCNT_2_@Co_3_O_4_; (**c**) MWCNT_3_@Co_3_O_4_; (**d**) MWCNT_4_@Co_3_O_4_.

**Figure 2 molecules-28-03177-f002:**
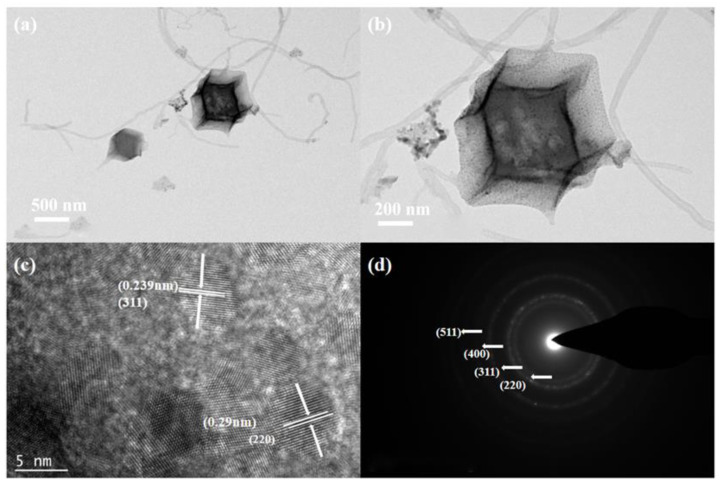
(**a**,**b**) TEM diagrams at different magnifications of MWCNT_3_@Co_3_O_4_; (**c**,**d**) high-resolution transmission electron microscopy (HRTEM) and selected area electron diffraction (SAED) of MWCNT_3_@Co_3_O_4_.

**Figure 3 molecules-28-03177-f003:**
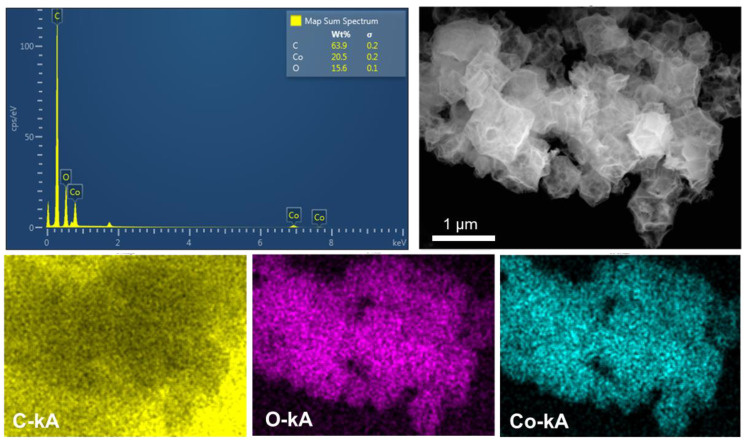
EDS analysis graph of MWCNT_3_@Co_3_O_4_.

**Figure 4 molecules-28-03177-f004:**
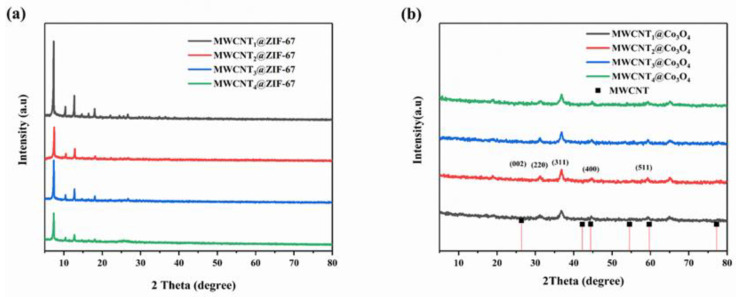
(**a**) XRD spectrum of MWCNTx@ZIF-67; (**b**) XRD spectrum of MWCNTx@Co_3_O_4_.

**Figure 5 molecules-28-03177-f005:**
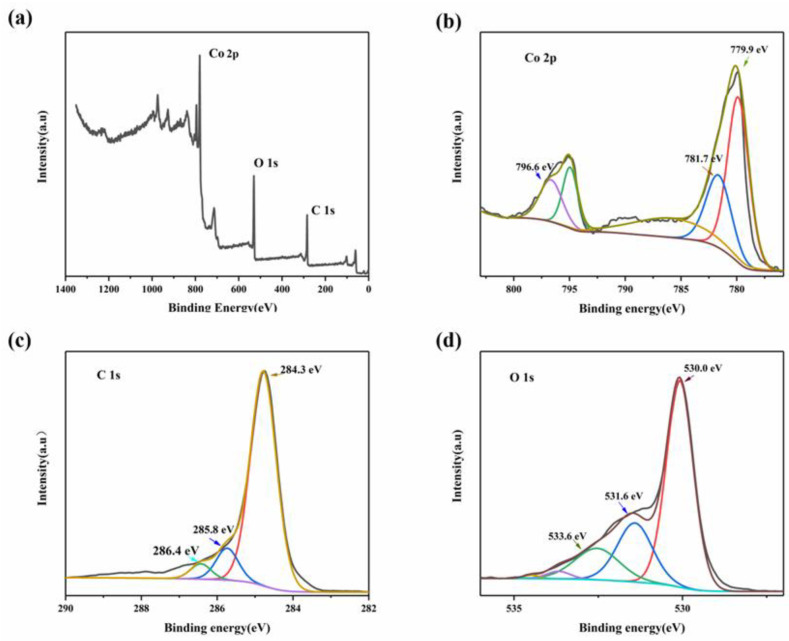
MWCNT3@Co_3_O_4_ (**a**) XPS full spectrum; (**b**) Co 2p; (**c**) C 1s and (**d**) O 1s XPS spectrum of fine spectrum.

**Figure 6 molecules-28-03177-f006:**
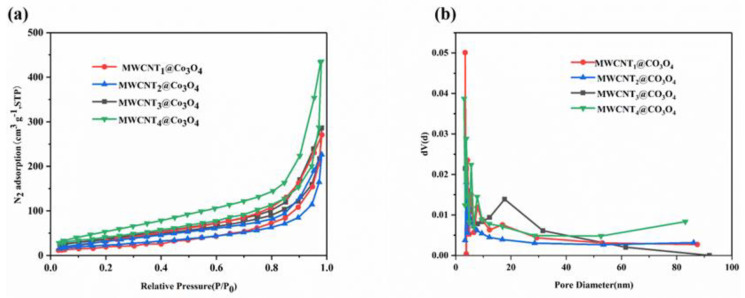
MWCNTx@Co_3_O_4_ composites (**a**) N_2_ adsorption desorption isotherm and (**b**) pore size distribution line.

**Figure 7 molecules-28-03177-f007:**
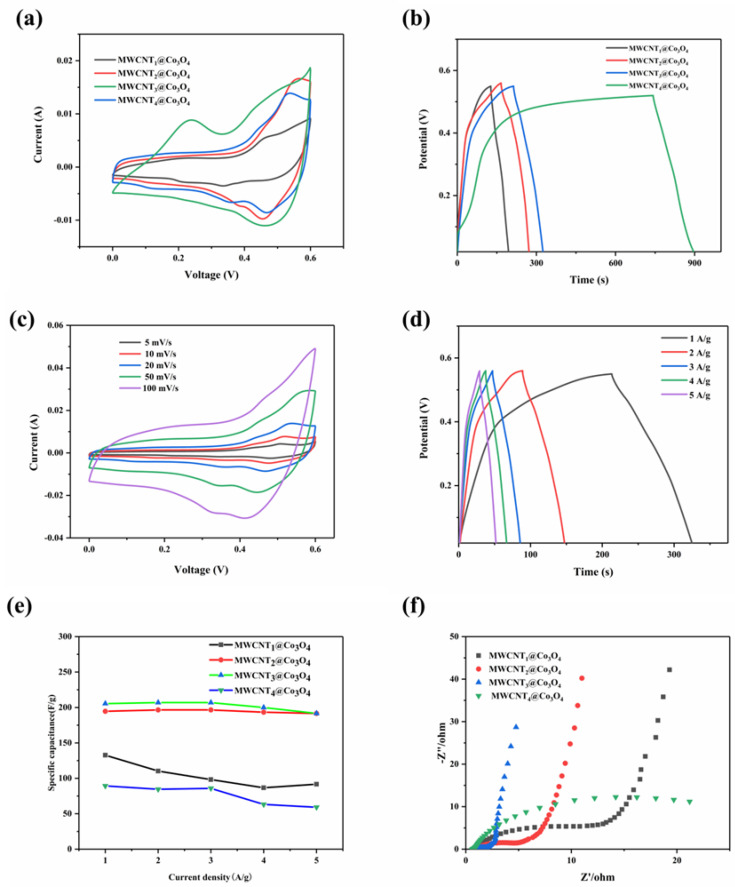
(**a**) CV curve of different mass ratios of MWCNT_3_@Co_3_O_4_ composites at 20 mV·s^−1^ sweep velocity; (**b**) GCD curve of the different mass ratios of MWCNT_3_@Co_3_O_4_ composites at 1A·g^−1^; (**c**) CV diagram of MWCNT_3_@Co_3_O_4_ composite at 20 mV·s^−1^ scanning speed; (**d**) GCD diagram of MWCNT_3_@Co_3_O_4_ composites at different current densities; (**e**) magnification performance diagram of the different mass ratios of MWCNT_3_@Co_3_O_4_ composites; (**f**) Nyquist diagram of the different mass ratios of MWCNT_3_@Co_3_O_4_ composites with an illustration of its local enlarged view.

**Figure 8 molecules-28-03177-f008:**
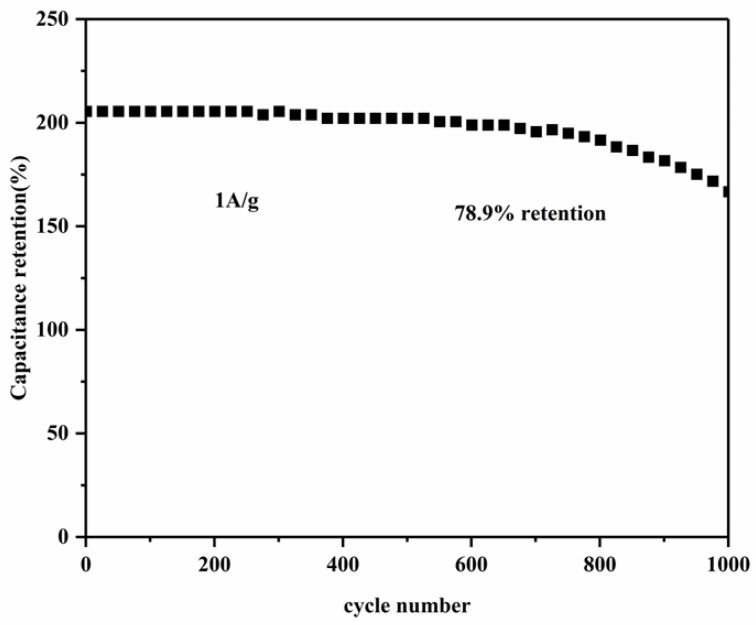
Cycles performance test of MWCNT_3_@Co_3_O_4_ at 1 A·g^−1^.

**Figure 9 molecules-28-03177-f009:**
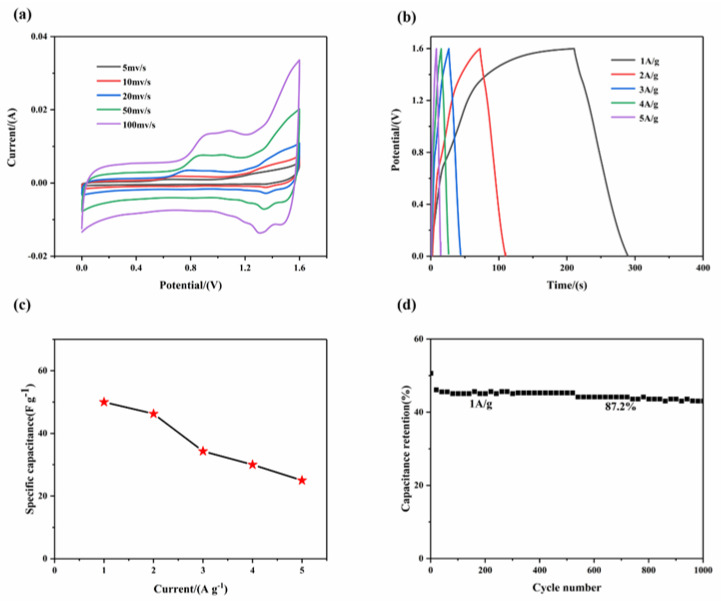
(**a**) Cyclic voltammetry curves at different sweep speeds; (**b**) charge–discharge curves under different current densities; (**c**) relation curve between current density and specific capacitance; (**d**) diagram of 1000 cycles under current density of MWCNT_3_@Co_3_O_4_//AC asymmetric supercapacitor (1 A·g^−1^).

**Figure 10 molecules-28-03177-f010:**
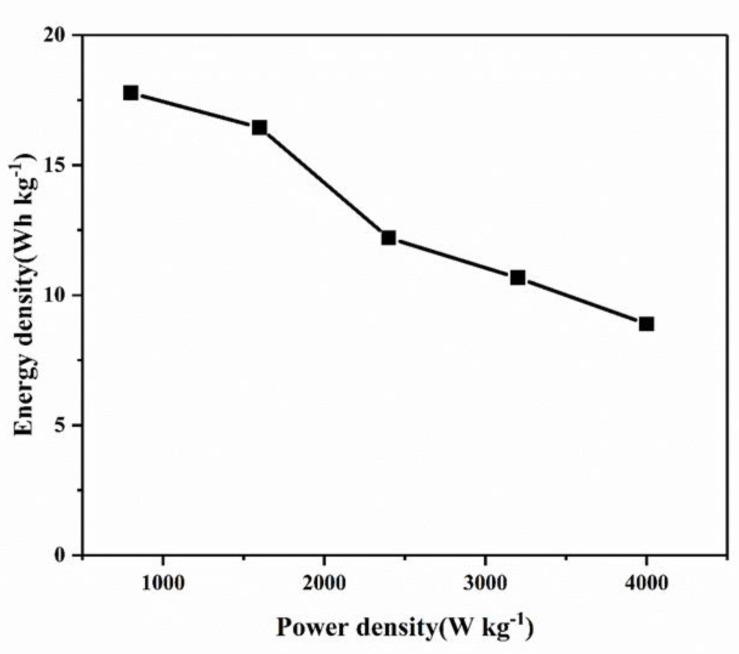
Ragone plots of the symmetrical supercapacitor.

**Table 1 molecules-28-03177-t001:** Element content analysis of different compound ratios of MWCNT@Co_3_O_4_.

Sample	Element Type and Content Wt%			Total Elements
	C	Co	O	
MWCNT_1_@Co_3_O_4_	21.23	50.65	28.12	100%
MWCNT_2_@Co_3_O_4_	26.07	44.04	29.90	100%
MWCNT_3_@Co_3_O_4_	27.50	45.20	27.30	100%
MWCNT_4_@Co_3_O_4_	28.50	44.32	27.18	100%

**Table 2 molecules-28-03177-t002:** The electrochemical performance of a summary of related Co_3_O_4_ materials.

Co_3_O_4_ Materials	Specific Capacitance (F·g^−1^)	Power Density (W·kg^−1^)	Energy Density (Wh·kg^−1^)	Ref.
rGO-Co_3_O_4_	472 F·g^−1^ (2 mV·s^−1^)	39.0	8.3	[44]
Nanostructured Co_3_O_4_	162 F·g^−1^ (2.75 A·g^−1^)	----	----	[45]
Co_3_O_4_ nanotubes	574 F·g^−1^ (0.1 A·g^−1^)	----	----	[46]
Co_3_O_4_/graphene	362.6 F·g^−1^ (0.72 A·g^−1^)	----	----	[47]
Mn_0.05_Co_2.95_O_4_	80.8 F·g^−1^ (1 A·g^−1^)	----	----	[48]
Co_3_O_4_/rGO/NF	311 F·g^−1^ (1 A·g^−1^)	12	40	[49]
MWCNT_3_@Co_3_O_4_	206.89 F·g^−1^ (1 A·g^−1^)	800	17.78	This work

## Data Availability

Not applicable.

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
