# Peer review of "Preparation of MoFs-Derived Cobalt Oxide/Carbon Nanotubes Composites for High-Performance Asymmetric Supercapacitor"

_molecules, 2023, doi:10.3390/molecules28073177_

Round 1
Reviewer 1 Report
MWCNTx@ZIF-67 is calcined to form MWCNTx@Co3O4 and applied to asymmetric supercapacitor. The topic of this manuscript is interesting. However, major revisions are required and the comments are given below.
1. The “cobalt” in the title is suggested to be replaced by “cobalt oxide”.
2. The capacitance performance of as synthesized materials is much smaller compared to many reports. Please give some explanations.
3. Transition metal compounds, porous carbon materials and their composites are widely applied in supercapacitors. Many researches have been done on this field. More references are suggested to be cited to enrich the content, e.g. Journal of Bioresources and Bioproducts 2022, 7 (1), 63-72; Journal of Bioresources and Bioproducts 2021, 6 (2), 142-151; Inorganic Chemistry Frontiers 2022, 9, 6108-6123.
4. Please add the research background in the abstract section, and do not simply list the various experimental characterization techniques and their results.
2. When the abbreviation is used for the first time, the full name should be given, and there is no need to give the full name for the subsequent use of the abbreviation such as the "EDLCs" and "AC" mentioned in the introduction.
3. “Fabrication of a supercapacitor electrode” is missing in the experimental section! Make accurate description of the experimental procedure so that others can repeat the experiment. For special reagents and instruments, the manufacturer, model and parameters shall be indicated.
4. The typesetting of the picture of the article is not correct, please modify according to the requirements of the journal.
5. A space should be added between the 500nm number and the letter in Figure 1; the font on Figure 2 ruler should be consistent with Figure 1, and the irrelevant number in Figure 2a should be cleared; with incorrect um format in Figure 3, it should be changed to μm; "C-kA" is not clear, so the brightness or color should be adjusted.
6. The resolution of figures is too poor (e.g. Figure 4-6). Please replace them with better ones.
7. Please pay attention to the writing of units. For example, the units should be written in the same style, “A/g” and “A g-1” need to be revised.
8. In line 213-214, the author is requested to carefully check whether the use of symbols in the chemical equation is consistent and meets the standards.
9. Please add nitrogen adsorption/desorption isotherms and pore size distribution curves.
10. There are some grammatical and punctuation errors in this manuscript:for example, C1s in Figure 5a needs to be modified; punctuation marks should uniformly use English punctuation marks, such as ";" in line 181, etc.
11. The author needs to analyze the results one by one, describe the differences in the series of samples, and put forward the reasons for the differences. For example, why is the GCD curve of sample MWCNT4@Co3O4 in Figure 6b asymmetrical? Please explain.
12. In the summary part, the author needs to point out the influence of experimental variables on the microstructure and performance, and give appropriate prospects to the materials and methods.
Author Response
MWCNTx@ZIF-67 is calcined to form MWCNTx@Co3O4 and applied to asymmetric supercapacitor. The topic of this manuscript is interesting. However, major revisions are required and the comments are given below.
Response: We appreciated your comments and suggestions that allowed us to further improve the quality of the present work. As can be found in the revised version of the manuscript, all the comments and suggestions have been strictly followed during the revising process. We sincerely hope to receive your understanding.
- The “cobalt” in the title is suggested to be replaced by “cobalt oxide”.
Answer. 1: Thank you very much for your kind suggestion. Your suggestion is very helpful to improve the quality of our paper. The title has been revised according to the opinion in revised manuscript.
- The capacitance performance of as synthesized materials is much smaller compared to many reports. Please give some explanations.
Answer. 2: Thank you very much for your kind suggestion. Your suggestion is very helpful to improve the quality of our paper. This work designs diamond dodecahedron structure of MWCNTx@Co3O4 composites. The specific capacitance of the MWCNT3@Co3O4 composite is 206.89 F g-1. Compared with high performance materials, the MWCNTx@Co3O4 composites may be larger in size, and the ratio of composites is not appropriate. We will optimize it in the future work, and hope to get your understanding.
- Transition metal compounds, porous carbon materials and their composites are widely applied in supercapacitors. Many researches have been done on this field. More references are suggested to be cited to enrich the content, e.g. Journal of Bioresources and Bioproducts 2022, 7 (1), 63-72; Journal of Bioresources and Bioproducts 2021, 6 (2), 142-151; Inorganic Chemistry Frontiers 2022, 9, 6108-6123.
Answer 3. Thank you very much for your kind suggestion and correction. Your suggestion is very helpful to improve the quality of our paper. More references have been cited to enrich the content e.g. Journal of Bioresources and Bioproducts 2022, 7 (1), 63-72; Journal of Bioresources and Bioproducts 2021, 6 (2), 142-151; Inorganic Chemistry Frontiers 2022, 9, 6108-6123 at [17], [35], [36].
- Please add the research background in the abstract section, and do not simply list the various experimental characterization techniques and their results.
Answer. 4: Thank you very much for your kind suggestion. Your suggestion is very helpful to improve the quality of our paper. We have added the research background in the abstract section, which has been highlight in red.
- When the abbreviation is used for the first time, the full name should be given, and there is no need to give the full name for the subsequent use of the abbreviation such as the "EDLCs" and "AC" mentioned in the introduction.
Answer. 5: Thank you very much for your kind suggestion. Your suggestion is very helpful to improve the quality of our paper. According to your suggestion, I have corrected the relevant content.
- “Fabrication of a supercapacitor electrode” is missing in the experimental section! Make accurate description of the experimental procedure so that others can repeat the experiment. For special reagents and instruments, the manufacturer, model and parameters shall be indicated.
Answer. 6: Thank you very much for your kind suggestion. Your suggestion is very helpful to improve the quality of our paper. According to your suggestion, I have added the relevant content.
2.4 Fabrication of supercapacitor electrode
(1) Preparation of single electrode
All single electrodes in this paper were prepared in the following way: firstly, a certain amount of active substance and acetylene black were weighed, dissolved in 5 ml of absolute ethanol, and ultrasonicated for 30 min, then a certain amount of PTFE lotion (the mass ratio of active substance, acetylene black and PTFE lotion (5wt%) was 8:1:1) was dropped in proportion, and then ultrasonicated for 10 min, and then dried in a constant temperature blast drying oven at 60 ℃ for 12 h. Finally, scrape the obtained black sample onto one side of foam nickel of 1 × 1 cm, press it with a tablet press and compact it evenly. By weighing the total mass of foam nickel before and after scraping, the total mass of active substance can be calculated.
(2) Assembly of Asymmetric Supercapacitors
In order to evaluate the practical application of the prepared electrode material, the electrode material and AC prepared in this article were used to prepare a single electrode in the same manner as in (1). Subsequently, when assembling the supercapacitor, they were assembled on the positive electrode and the negative electrode of the supercapacitor, respectively. At the same time, the positive and negative electrodes were separated by a membrane, and an asymmetric supercapacitor was assembled. The electrolyte was prepared as a 3.0 M KOH solution”.
- The typesetting of the picture of the article is not correct, please modify according to the requirements of the journal.
Answer. 7: Thank you very much for your kind suggestion. Your suggestion is very helpful to improve the quality of our paper. According to your suggestion, I have revised the relevant picture.
- A space should be added between the 500nm number and the letter in Figure 1; the font on Figure 2 ruler should be consistent with Figure 1, and the irrelevant number in Figure 2a should be cleared; with incorrect um format in Figure 3, it should be changed to μm; "C-kA" is not clear, so the brightness or color should be adjusted.
Answer. 8: Thank you very much for your kind suggestion. Your suggestion is very helpful to improve the quality of our paper. According to your suggestion, I have revised the relevant content. The A space has been added between the 400nm number and the letter in Figure 1(a-d); the font on Figure 2 ruler has been consistent with Figure 1, and the irrelevant number in Figure 2a has been cleared; with incorrect um format in Figure 3, i has been changed to μm; The color of "C-kA, Co-KA and O-KA" in Figure 3 has been adjusted.
- The resolution of figures is too poor (e.g. Figure 4-6). Please replace them with better ones.
Answer. 9: Thank you very much for your kind suggestion. Your suggestion is very helpful to improve the quality of our paper. According to your suggestion, I have revised the relevant content.
- Please pay attention to the writing of units. For example, the units should be written in the same style, “A/g” and “A g-1” need to be revised.
Answer. 10: Thank you very much for your kind suggestion. Your suggestion is very helpful to improve the quality of our paper. According to your suggestion, I have revised the relevant content.
- In line 213-214, the author is requested to carefully check whether the use of symbols in the chemical equation is consistent and meets the standards.
Answer. 11: Thank you very much for your kind suggestion. Your suggestion is very helpful to improve the quality of our paper. The use of symbols in the chemical equation has been checked carefully and it is consistent and meets the standards.
- Please add nitrogen adsorption/desorption isotherms and pore size distribution curves.
Answer. 12: Thank you very much for your kind suggestion. Your suggestion is very helpful to improve the quality of our paper. The nitrogen adsorption/desorption isotherms and pore size distribution curves has been added in Figure. 6 and the responding word description is added. The added content is as follows.
Figure 6. MWCNTx@Co3O4 composites (a) N2 adsorption desorption isotherm and (b) pore size distribution line.
Figure. 6 shows the pore structure and specific surface area of the MWCNTx@Co3O4 composites. As can be seen from Figure. 6.(a), The adsorption/desorption isotherm of MWCNTx@Co3O4 slowly rises at first, and when it reaches a certain pressure, it rises sharply, resulting in a vertical tail ring and a hysteresis ring. This feature indicates that MWCNTx@Co3O4 composite materials have microporous, mesoporous, and macroporous structures, mainly mesoporous. The specific surface areas of MWCNT1@Co3O4, MWCNT2@Co3O4, MWCNT3@Co3O4 and MWCNT4@Co3O4 are 104.20, 66.44, 99.56 and 136.97 m2 g-1, respectively. In addition, the pore diameter distribution curve of MWCNTx@Co3O4 is shown in Figure. 6(b), The pore volumes of MWCNT1@Co3O4, MWCNT2@Co3O4, MWCNT3@Co3O4 and MWCNT4@Co3O4 were 3.39, 3.79, 3.06, and 3.30 nm, respectively. It can be well seen in the figure, The pore size distribution of MWCNT3@Co3O4 is mainly mesoporous, mainly due to the insertion of MWCNTs into Co3O4, resulting in a larger pore size of the overall material. What’s more, the mesoporous structure is conducive to ion diffusion, thereby improving the electrochemical performance of the material. Therefore, the obtained electrode material should have good electrochemical performance.
- There are some grammatical and punctuation errors in this manuscript:for example, C1s in Figure 5a needs to be modified; punctuation marks should uniformly use English punctuation marks, such as ";" in line 181, etc.
Answer. 13: Thank you very much for your kind suggestion. Your suggestion is very helpful to improve the quality of our paper. According to your suggestion, C1s in Figure 5a has been modified to C 1s; The punctuation marks of ";" has been modified to ‘,’
- The author needs to analyze the results one by one, describe the differences in the series of samples, and put forward the reasons for the differences. For example, why is the GCD curve of sample MWCNT4@Co3O4 in Figure 6b asymmetrical? Please explain.
Answer. 14: Thank you very much for your kind suggestion. Your suggestion is very helpful to improve the quality of our paper. The difference has been described and put forward the reasons for the differences. “It is observed that the curve is almost triangular in symmetry, and there is no particularly rapid drop in voltage, indicating that the synthesized MWCNT1@Co3O4, MWCNT2@ Co3O4, MWCNT3@ Co3O4 materials have particularly good electrochemical reversibility. While the GCD curves of the MWCNT4@ Co3O4 composite is different from others, its charge time is longer from its discharge time. It is mainly due to the higher resistance which is consistent to the result of Figure. 7 (f). When apply the same voltage in the MWCNT4@ Co3O4 composite, the current is unable to charge fastly, which leads to the GCD curve of sample MWCNT4@ Co3O4 composite asymmetrical”.
- In the summary part, the author needs to point out the influence of experimental variables on the microstructure and performance, and give appropriate prospects to the materials and methods.
Answer. 15: Thank you very much for your kind suggestion. Your suggestion is very helpful to improve the quality of our paper. According to your suggestion, I have revised the relevant content.
In summary, the diamond dodecahedron structure MWCNTx@Co3O4 composite has been successfully fabricated by a smart approach. With the increase of MWCNT content, the electrochemical properties of the MWCNTX@Co3O4 composite first increased and then decreased. Benefiting from the unique structure, high specific surface area and reasonable pore size pore size distribution, the as-obtained MWCNT3@Co3O4 composite exhibits satisfactory capacitive behavior: 206.89 F g-1 at a current density of 1 A g-1; excellent cycling stability of 87.2% capacitance retention over continuous 1000 cycles. An asymmetric supercapacitor cell was fabricated through MWCNTx@Co3O4 and AC as positive and negative electrode respectively. The cell can deliver a high energy density of 17.78 Wh kg-1 at a power density of 800 W kg-1. Our results suggest that the MWCNTx@Co3O4 composite can be used in actual high power devices. The preparation strategy offers a facile and variable route to rational design and prepare cobalt oxide electrode materials for a variety of applications in energy storage and conversion, catalysis, and environmental treatment.
Reviewer 2 Report
1. The resolution of many graphs is so poor that they look blurry when zoomed in. Among them, the fonts of the lattice plane and interlayer d spacing in Figure 2 are almost invisible
2. The English abbreviations appearing in the abstract should clearly indicate his original full name.
3. The nomenclature in the preparation method is very confusing, too many numbers will confuse the reader. And is solution A mixed with solution B afterwards? The author does not seem to state.
4. I am curious how big of the measuring area?
5. There have been a lot of related researches in recent years, please cite appropriate literature, such as: Electrochimica Acta 403 (2022) 139692 and Journal of the Taiwan Institute of Chemical Engineers 134 (2022) 104318.
Author Response
- The resolution of many graphs is so poor that they look blurry when zoomed in. Among them, the fonts of the lattice plane and interlayer d spacing in Figure 2 are almost invisible
Answer. 1: Thank you very much for your kind suggestion. Your suggestion is very helpful to improve the quality of our paper. According to your suggestion, the figure 2 has been revised and the fonts of the lattice plane and interlayer d spacing in Figure 2 are visible. Many graphs have been changed completely.
- The English abbreviations appearing in the abstract should clearly indicate his original full name.
Answer. 2: Thank you very much for your kind suggestion. Your suggestion is very helpful to improve the quality of our paper. According to your suggestion, I have revised the relevant content. The SEM and TEM characterization in abstract has been revised to scanning electron microscope (SEM) and Transmission electron microscope (TEM), The XRD and XPS has been revised to X-ray diffraction (XRD) and X-ray Photoelectron Spectroscopy (XPS) The AC has been revised to activated carbon (AC).
- The nomenclature in the preparation method is very confusing, too many numbers will confuse the reader. And is solution A mixed with solution B afterwards? The author does not seem to state.
Answer. 3: Thank you very much for your kind suggestion. Your suggestion is very helpful to improve the quality of our paper. According to your suggestion, I have revised the relevant content.
Preparation of MWCNTx@ZIF-67 precursor: First, different mass of MWCNT pour into polyethylene pyrrolidone and methanol solution, ultrasonic 30 min, and then add a certain amount of cobalt nitrate, magnetic stirring for 10 min. In addition, 2-methylimidazole was added to the above solution stirring for 1 h, and then the prepared solution was left to stand at room temperature for 24h. After centrifugation and wash, the black precipitate was dried at 60℃ for 12 h, and the precursor MWCNTx@ZIF-67 was obtained. The weights of MWCNT were 0.0203g, 0.0415g, 0.0608g and 0.1510g, respectively. Therefore, the resultant MWCNT@ZIF-67 numbers are MWCNT1@ZIF-67, MWCNT2@ZIF-67, MWCNT3@ZIF-67 and MWCNT4@ZIF-67, respectively.
Preparation of MWCNTx@Co3O4 composite: The synthesized precursors (MWCNT1@ZIF-67, MWCNT2@ZIF-67 and MWCNT3@ZIF-67, MWCNT4@ZIF-67) were annealed in air at 350 °C for 2 hours at a heating rate of 4 ℃·min-1. The obtained samples were labeled MWCNT1@Co3O4, MWCNT2@Co3O4, MWCNT3@Co3O4 and MWCNT4@Co3O4.
- I am curious how big of the measuring area?
Answer. 4: Thank you very much for your kind suggestion. Your suggestion is very helpful to improve the quality of our paper. I have added the relevant content, and the measuring area is 1×1cm2.
- There have been a lot of related researches in recent years, please cite appropriate literature, such as: Electrochimica Acta 403 (2022) 139692 and Journal of the Taiwan Institute of Chemical Engineers 134 (2022) 104318.
Answer 5. Thank you very much for your kind suggestion and correction. Your suggestion is very helpful to improve the quality of our paper. Appropriate literatures have been cited such as Electrochimica Acta 403 (2022) 139692 and Journal of the Taiwan Institute of Chemical Engineers 134 (2022) 104318 at [18], [20].

Reviewer 3 Report
The present manuscript details a straightforward and innovative approach to producing Cobalt oxide/carbon nanotube composites for use in supercapacitors. The experiments and findings are thoroughly explained. It is recommended that the author provide further analysis on why the C/Co mass ratio impacts device performance, based on structure and theory.
Numerous high-performance materials and composites for supercapacitors have been reported in literature. To contextualize the significance of this MOF-derived composite, it would be useful for the authors to include a summary table of existing materials and where this composite stands in comparison.
Author Response
The present manuscript details a straightforward and innovative approach to producing Cobalt oxide/carbon nanotube composites for use in supercapacitors. The experiments and findings are thoroughly explained. It is recommended that the author provide further analysis on why the C/Co mass ratio impacts device performance, based on structure and theory.
Answer 1. Thank you very much for your kind suggestion and correction. Your suggestion is very helpful to improve the quality of our paper. The C/Co mass ratio is analyzed by the EDS analysis of the composites.
Table 1 shows the composite ratio of MWCNT@Co3O4. As can be seen from the table, with the increase in the amount of MWCNT added, the content of C element in the composites gradually increased from 21.23% to 28.50%, while the content of Co element in the composites gradually decreased from 50.65% to 44.32%. This is confirmed that the addition of MWCNT. Through the above analysis, it is confirmed that the successful synthesis of the MWCNTx@Co3O4 composite materials.
Sample |
Element type and content Wt%
|
Total Elements |
||
C |
Co |
O |
||
MWCNT1@Co3O4 |
21.23 |
50.65 |
28.12 |
100% |
MWCNT2@Co3O4 |
26.07 |
44.04 |
29.90 |
100% |
MWCNT3@Co3O4 |
27.50 |
45.20 |
27.30 |
100% |
MWCNT4@Co3O4 |
28.50 |
44.32 |
27.18 |
100% |
Table1 Element content analysis of different compound ratio of MWCNT@Co3O4.
Numerous high-performance materials and composites for supercapacitors have been reported in literature. To contextualize the significance of this MOF-derived composite, it would be useful for the authors to include a summary table of existing materials and where this composite stands in comparison.
Answer 2. Thank you very much for your kind suggestion and correction. Your suggestion is very helpful to improve the quality of our paper. A summary table of existing materials has been listed as follows:
Table 2 The electrochemical performance of a summary of related Co3O4 materials.
Co3O4 materials |
Specific capacitance (F g-1) |
Power density (W kg-1) |
Energy density (W h Kg-1) |
Ref |
Co3O4@C |
251 F g-1 ( 1 A g-1) |
---- |
---- |
[40] |
rGO-Co3O4 |
472 F g−1( 2 mV s−1 ) |
39.0 |
8.3 |
[45] |
Nanostructured Co3O4 |
162 F g−1 (2.75 A g-1) |
---- |
---- |
[46] |
Co3O4 nanotubes |
574 F g−1 ( 0.1A g-1) |
---- |
---- |
[47] |
Co3O4/graphene |
362.6 F g-1 (0.72A g-1) |
---- |
---- |
[48] |
Mn0.05Co2.95O4 |
80.8 F g-1 (1A g-1) |
---- |
---- |
[49] |
Co3O4/rGO/NF |
311 F g-1 ( 1 A g-1) |
12 |
40 |
[50] |
MWCNT3@Co3O4 |
206.89 F g-1 (1 A g-1) |
800 |
17.78 |
This work |

Round 2
Reviewer 1 Report
The manuscript has been revised and could be accepted.